# Evaluation of Ultra–High–Pressure Sterilization in Terms of Bactericidal Effect, Qualities, and Shelf Life of ‘Xinli No. 7’ (*Pyrus sinkiangensis*) Pear Juice

**DOI:** 10.3390/foods12142729

**Published:** 2023-07-18

**Authors:** Xiaojing Gan, Zhizhou Chen, Liwen Wang, Wenhui Liu, Qianyun Ma, Rongbin Li, Jie Wang, Jianlou Mu

**Affiliations:** 1College of Food Science and Technology, Hebei Agricultural University, Baoding 071000, China; gxj0375@163.com (X.G.); liwenwang.claire@outlook.com (L.W.); maqianyun@126.com (Q.M.); lirongbin77@126.com (R.L.); wj591010@163.com (J.W.); 2College of Mechanical and Electrical Engineering, Hebei Agricultural University, Baoding 071000, China; chenzhizhou2003@126.com; 3Beijing Huiyuan Food and Beverage Co., Ltd., Beijing 101314, China; lwenhui2004@126.com

**Keywords:** shelf life, pear juice, ultra–high pressure, sterilization, storage

## Abstract

In this study, ultra–high–pressure sterilization (UHPS) of Xinli No. 7 juice (XL7) was explored and optimized. A challenge to implement UHPS in juice as a full alternative to thermal treatment could be represented by the adoption of a pressure level of up to 500 MPa for 20 min at one cycle followed by the packaging in aseptic conditions. It was found that UHPS and HS treatments could effectively kill the microorganisms in XL7 juice but HS treatment would inevitably lose the nutritional quality in the juice, while UHPS treatment could better maintain the glyconic acid content, functional components, and antioxidant activity and reduce Browning degree and improve the stability of XL7 juice. The deterioration rate of UHPS and HS–treated XL7 juice increased with the increased storage temperature. The predicted shelf life of UHPS and HS–treated XL7 juice was 68 and 41 days at 4 °C, respectively. Collectively, UHPS treatment combined with low–temperature storage might be an effective way to prolong the shelf life of XL7 juice.

## 1. Introduction

Pear is known as “the family of hundreds of fruits”, with rich nutritional components and medicinal value [1]. NFC (not from concentrate) pear juice can maintain the nutritional value and sensory quality of the pear and has great prospects for development [2]. However, NFC pear juice application studies are few and some issues related to the production and maintenance of pear juice still exist [3]. For example, browning occurs easily during the preparation of NFC pear juice [4] and excessive addition might reduce the nutritional value of NFC pear juice [5]. Therefore, interest in minimum processing techniques, including ultra–high pressure (UHP), pulsed electric field, and ultrasound processing, has increased recently, among which UHP has been effectively used for a variety of fruit–based products such as juices and smoothies [6]. Considering that flavour is frequently cited as the most important characteristic of fruit–based products by consumers [7], it is essential to evaluate the sensory quality and physicochemical properties of UHP–treated juices before they are introduced to the market.

Conventional thermal processing has been the principal method for processing juices for a long time, with typical thermal treatment using heat treatments between 60 and 100 °C to destroy target microorganisms or enzymes. This can cause undesirable changes [8]. Yet, it is recognized that the loss of volatile compounds during heating processing may lead to a “cooked” taste [9]. NFC pear juice is prone to microbial contamination during preparation [10]. This conventional heat–sterilization method can kill microorganisms and may affect the flavour of juice [11]. The growth and propagation of microorganisms is the root cause of juice spoilage [12]. Hence, it is necessary to adopt effective treatments to ensure the edible safety of juice. UHP can achieve sterilization at a lower temperature without heating juice. UHP treatment can not only address the limitations related to thermal treatment but also reduce the energy consumption typically associated with heating and cooling. UHP sterilization (UHPS) utilizes a certain degree (above 140 °C) of static liquid pressure to the packaged juice to kill microorganisms while maintaining the quality of the juice [13]. Niu, et al. [14] studied the effects of UHPS and heat–sterilisation (HS) treatments on the nutritional quality and sensory properties of passion fruit juice, finding that the UHPS–treated passion fruit juice could maintain its good sensory quality without damaging its nutritional properties.

Furthermore, the food industry has highlighted the necessity of tailor–made protocols to maximize the shelf life of UHPS and HS–treated products without detrimental effects on the nutritional functionality. The shelf life of juice refers to the time when the juice after sterilization and storage is safe and harmless under appropriate conditions and its sensory, physical, chemical, and microbial indexes meet the requirements of consumers, as well as its nutritional substances meet the requirements [15]. Shelf life is a key factor restricting the production and economic benefits of fruit juice, which has attracted much attention from the juice market. Therefore, the prediction of juice shelf life is particularly important. Shelf life is determined by product processing technology, nutritional quality, and microbial content [16]. The sensory prediction model was built under the guidance of the principle of accelerated shelf–life testing [17]. NFC pear juice can retain the fresh flavour of pear fruit. However, its shelf life is generally low, which restricts the production of NFC pear juice.

Few studies focused on nonthermal sterilization treatment with NPC pear juice as well as its effect on the quality during storage. In our preliminary study [18], 16 varieties of pears were used and their sensory quality, physicochemical characteristics and volatile components of NFC pear juice prepared by these peers were evaluated to determine the most suitable pear varieties for NFC juice preparation. According to the results, the Xinli No. 7 (XL7) pear was finally selected as the raw material in this study. The effect of UHPS treatment on the microbiological quality of XL7 juice was explored, and the quality of XL7 juice prepared by UHPS and HS treatments was compared to determine the optimized sterilization method of XL7 juice. In addition, the quality changes of XL7 juice treated with UHPS and HS during storage were studied and the shelf life of UHPS– and HS–treated juice was also predicted to determine the optimized conditions for storage to extend the shelf life of XL7 juice.

## 2. Materials and Methods

### 2.1. Materials

Xinli No. 7 (XL7) specimens were purchased from a supermarket in Weixian, Hebei Province when they reached the maturity of the respective commercial harvesting period and immediately stored at 4 °C after sampling for subsequent processing analysis. The fruits with diseases, mechanical damage, and immaturity were removed. Hydrochloric acid, sulfuric acid, methanol, anhydrous ethanol, ethyl acetate, zinc acetate, sodium carbonate, sodium bicarbonate, sodium nitrite, aluminium nitrate, sodium chloride, sodium hydroxide (analytically pure), sucrose, potassium ferricyanide, phenolphthalein, polyethylene glycol (PEG 6000, analytically pure), acetic acid–sodium acetate buffer solution (0.1 mol/L, pH = 5.5), Triton X–100, catechol, and potassium persulfate were purchased from Sinopharm Chemical Reagent Co., Ltd. (Beijing, China). Folinol (1 mol/L), gallic acid, rutin, 1,1-diphenyl-2-trinitrophenylhydrazine (DPPH), and 2,2′-diazo-di-3-ethylbenzothiazolin-6-sulfonic acid (ABTS) were bought from Soleibao Biotechnology Co., Ltd. (Shanghai, China).

### 2.2. Operation of Ultrasonic–Ascorbic Acid Compound Colour Protection Technology

Six ‘Xinli No. 7’ (XL7) were cleaned, peeled, and cut into pieces to soak them in 500 mL of 0.18% ascorbic acid solution for 20 s and then immediately pressed into XL7 juice. The juice was filtered through an 80–mesh sieve. A total of 500 mL of filtrate was placed in a beaker with an ice bath for ultrasonic treatment. The horn diameter of the ultrasonic device was 18 mm and the probe of the horn was extended 20 mm below the pear juice level. The ratio of the ultrasonic start time to the ultrasonic pause time was 2:3. The single–factor test was carried out by adjusting the condition, as shown in Appendix A. The ultrasonic power was set as 300 W for 10 min according to our preliminary study results [18].

### 2.3. Optimization of XL7 Juice by Ultra–High–Pressure Sterilization (UHPS) Processing

#### 2.3.1. Single Factor Test

A total of 50 mL of XL7 juice obtained in optimized colour–protection conditions was immediately filled into a double sterile polyethylene bag (110 mm × 180 mm) for further analysis. XL7 juice was put into the pressure chamber with water as the pressure transmission medium, and the single–factor test was carried out by adjusting the parameters of the ultra–high–pressure equipment.

#### 2.3.2. Effect of Processing Parameters on the Number of XL7 Juice Colonies

The pressure holding time was set as 15 min; the temperature was controlled at 20 °C. XL7 juice was pressurized once and was treated under pressures of 100, 200, 300, 400, and 500 Mpa separately.

The pressure was set at 300 Mpa; the temperature was controlled at 20 °C. XL7 juice was pressurized once and was treated at 5, 10, 15, 20, and 25 min, separately.

The pressure was set to 300 Mpa, and the holding time was set to 15 min. The temperature was controlled to 20 °C. The cycles of 1, 2, 3, 4, and 5 were adjusted separately.

The pressure was set at 300 Mpa and the holding time was set at 15 min, XL7 juice was processed once, and the temperatures were 10, 20, 30, 40 s and 50 °C, separately.

The juice was cooled in an ice bath at 0 °C immediately, and the total number of colonies in the juice was measured separately after these treatments.

#### 2.3.3. Orthogonal Optimization Test

According to the results of the single–factor test, the test temperature was controlled at 20 °C and three factors, including pressure (A), pressure holding time (B) and cycle times (C) were selected for the orthogonal optimization test (Table 1). The total number of colonies was taken as the index to determine the appropriate ultra–high–pressure sterilization conditions for XL7 juice.

#### 2.3.4. Influence of UHPS and HS Processing on Sterilization Efficiency and Juice Quality

UHPS: XL7 juice was selected to measure its microbial indexes (total colony count, mould and yeast, and *E. coli*); colour quality (colour difference, and Browning degree, soluble quinone, and 5–HMF); sugar–acid content (soluble solids, total acid, and their ratio); stability quality (centrifugal precipitation rate) and functional characters (polyphenols, flavonoids, DPPH, and ABTS radical scavenging abilities) in a 0 °C ice bath. The methods of these indexes are described in the following sections. Nonsterilized (NS) juice was set as the control group.

HS: A total of 50 mL of XL7 juice obtained in optimized colour–protection conditions was immediately filled into a double sterile polyethylene bag (110 mm × 180 mm), sealed, and cooled in an ice bath at 0 °C.

### 2.4. Determination of Colour Difference

The XL7 juice sample was put into the transparent container, and the L*, a*, and b* values of the juice were measured by the colour–difference metre. The colour difference (∆E) was calculated as the following Equation (1) [19]:(1)$$∆E=\sqrt{{\left(L*-L_{0}*\right)}^{2}+{\left(a*-a_{0}*\right)}^{2}{+\left(b*-b_{0}*\right)}^{2}}$$
where L_0_, a_0_, and b_0_ were the chromatic values of fresh juice, while L, a, and b were the chromatic values of juice after colour protection by ultrasonic technique (conducted in our preliminary study).

### 2.5. Determination of Browning Degree and Browning Inhibition Rate (BIR)

The browning degree was determined by referring to the method of a previous study [20] at 420 nm. The BIR can be calculated by the following Formula (2):(2)$$\mathrm{BIR}=\left(1-\frac{A_{i}}{A_{0}}\right)\times 100$$
where $A_{0}$ and $A_{i}$ represents the Browning degree of the fresh juice and colour–protected juice, respectively.

### 2.6. Determination of Polyphenol Oxidase (PPO) Activity

A total of 10 mL of XL7 juice was mixed with 10 mL of extract (containing 0.1 mol/L of 4% PVPP, 0.34% PEG 6000 and 1% Triton X–100, acetate buffer (pH = 5.5)) at 4 °C. After centrifugation at 12,000× *g* for 30 min, 100 μL of supernatant was mixed with the substrate (1 mL of 0.05 mol/L catechol and 4 mL of 0.05 mol/L acetate buffer, pH = 5.5), up to 5mL. The absorbance at 420 nm was measured every 1 min for 6 min. PPO activity is calculated according to the following Formula (3) [21]:(3)$$U=\frac{{∆\mathrm{OD}}_{420}\times V}{V_{S}\times V_{Z}}$$
where U is the PPO activity (${∆\mathrm{OD}}_{420}/\min/\mathrm{mL}$); ${∆\mathrm{OD}}_{420}$ is the absorbance change of the reaction mixture per minute; V represents the total volume of sample extract; $V_{S}$ is the volume of supernatant, while $V_{Z}$ is the total volume of sample.

### 2.7. Determination of Soluble Quinone and 5–Hydroxymethylfurfural (5–HMF) Contents

Soluble quinone and 5–HMF contents were determined by referring to the method of previous studies [22,23]. Briefly, 0.5 g of the sample was mixed with 5 mL of 1% (*v*/*v*) HCl–methanol solution, and then the mixture was centrifuged at 4 °C at 8000 g for 10 min. The supernatant was collected, and the absorbance was measured at OD_437_. The soluble quinone content was expressed as OD_437_/g.

XL7 juice (5 g) was dissolved in 25 mL of distilled water. The diluted juice sample was mixed with 0.5 mL of 15% potassium ferrocyanide and 0.5 mL of 30% zinc acetate solution. Afterwards, 25 mL of the sample was aspirated and centrifuged at 9000 r/min for 10 min. The supernatant (5 mL) was collected into two test tubes, one of which was mixed with 0.2 g/100 mL of NaHSO_3_, and its absorbance was measured at 284 nm, while the other one was mixed with 5 mL of distilled water, and the absorbance at 336 nm was determined. The 5–HMF content was calculated according to the following formula:$$X_{5–\mathrm{HMF}}=\frac{\left({\mathrm{OD}}_{284}-{\mathrm{OD}}_{336}\right)\times 14.97\times 5}{m}$$
where X_5–HMF_ represents the 5–HMF content in XY7 juice (mg/100g); OD_284_ and OD_336_ are the absorbance value at 284 and 336 nm, respectively; and m is the mass of XL7 juice (g).

### 2.8. Evaluation of Glyconic Acid Content in XL7 Juice

The total soluble solids (TSS), titratable acid, soluble sugar, solid acid ratio, and sugar–acid ratio were determined according to previous studies [24].

### 2.9. Polyphenol and Flavonoid Contents in XL7 Juice

The polyphenol content was determined with reference to a study from [25]. The result was expressed as the equivalent of gallic acid/100 mL of XL7 juice (mg/100 mL). The flavonoid content was determined according to a study from Matute, et al. [26]. The result was expressed as the equivalent of rutin in 100 mL of XL7 juice (mg/100 mL).

### 2.10. Determination of DPPH and ABTS Free Radical Scavenging Ability

DPPH and ABTS free–radical scavenging abilities were determined according to the method of Loganayaki, et al. [27] The absorbance was measured at 517 and 734 nm, respectively. The standard curve equations for DPPH and ABTS were y = 1.6955x + 0.081 (r² = 0.9975) and y = −1.0278x + 9.6993 (r^2^ = 0.9955), respectively.

### 2.11. Identification of Volatile Components in XL7 Juice

A total of 2 mL of XL7 juice were placed in a 20 mL headspace bottle and the volatile components of pear juice were determined by the headspace–gas chromatography–ion migration spectrometry (HS–GC–IMS) technique [28]. The relative concentration of volatile components of XL7 juice was determined by the relative peak intensity. The conditions of the HS–GC–IMS instrument are shown in Appendix A.

### 2.12. Sensory Analysis of XL7 Juice

A total of 10 trained laboratory professionals were selected to form an evaluation team [29]. The sensory evaluation of XL7 was conducted from four aspects. The overall sensory evaluation score of XL7 juice was the average score of the four indicators, as shown in Appendix A.

### 2.13. Determination of Microbial Indexes in XL7 Juice

The CFU, mould and yeast, and *E. coli* were determined according to a previous study [30]. The results are expressed as log cfu/mL.

### 2.14. Effects of Different Sterilization Methods on Storage Quality and Shelf Life of XL7 Juice

The cooled XL7 juice obtained by UHPS and HS processing was stored at 4, 27, and 37 °C, separately, and the change–of–quality indexes, including microbial indexes (total colony count, mould and yeast, and *E. coli*); colour quality (colour difference, and Browning degree, soluble quinone, and 5–HMF); sugar–acid content (soluble solids, total acid, and their ratio); stability quality (centrifugal precipitation rate) and functional characters (polyphenols, flavonoids, DPPH, and ABTS radical scavenging abilities) during the storage period of the juice were monitored.

### 2.15. Accelerated Shelf–Life Test

The cooled XL7 juice obtained from Section 2.3.3 by UHPS and HS separately was stored at 4 (control), 27, and 37 °C, separately. Guided by the principle of the ALST method, the temperature–induced accelerated destruction test was used to predict the shelf life of the XL7 juice using the temperature–based Q10 model, according to the ALST method, which could be calculated according to Equation (4). The Q10 of canned food ranges from 1.1 to 4 [31]. Herein, the Q10 was set to be 2. The quality–related indexes described in Section 2.3.4 of the juice at 4, 27, and 37 °C were determined every 1, 2, and 4 days, respectively.
(4)$$Q_{10}=\frac{H_{\left(T\right)}}{H_{\left(T+10\right)}}$$
where $Q_{10}$ depicts the sensitivity of storage temperature to food reactions; $H_{\left(T\right)}$ is the shelf life at T °C; while $H_{\left(T+10\right)}$ represents the shelf life at T + 10 °C.

### 2.16. Storage Quality, Shelf Life and Volatile Compounds of XL7 Juice

The sensory quality, stability (centrifugal sediment rate, CSR), and the quality indexes described in Section 2.3.3 were determined to evaluate the storage quality of XL7 juice. A total of 30 mL XL7 juice was taken and centrifuged at 4 °C and 12,000 *g* for 10 min. The supernatant was discarded and the mass of sediment was recorded. The CSR was calculated according to Formula (5) [32]. Thereafter, according to the results, the shelf life was determined by the microbial and sensory score. The overall sensory score was lower than 5.5 and the microorganisms exceeded the standard as the test end points. The actual Q10 value of XL7 juice with two sterilization methods was calculated according to Formula (6):(5)$$\mathrm{CSR}=\frac{W_{1}}{W_{2}}\times 100$$
where CSR refers to the centrifugal sediment rate; $W_{1}$ is the weight of sediment; and $W_{2}$ is the XL7 juice before centrifugation.
(6)$$S_{T_{1}}={Q_{10}}^{\frac{∆T}{10}}\times S_{T_{2}}$$
where $S_{T_{1}}$ is the shelf life at *T*1; $Q_{10}$ depicts the sensitivity of storage temperature to food reactions; $S_{T_{2}}$ is the actual shelf life at *T*2; while ∆T represents the difference between the storage temperature of *T*1 and the test temperature *T*2.

The volatile components in the juice were determined on the 0 days without sterilization and before and after storage at 4, 27, and 37 °C after different UHPS and HS treatments, separately.

### 2.17. Statistical Analysis

Results were analysed using SPSS 26.0 software and expressed as a mean ± standard deviation (SD). *p* < 0.05 or *p* < 0.01 represents significant difference. The graphics are drawn using Origin 2019b software. The response surface test was designed and analysed by Design–Expert 8.0.6 software.

## 3. Results and Discussion

### 3.1. Effects of Different Sterilization Methods on Bactericidal Efficacy and Quality of XL7 Juice

As can be seen from Table 1 and Table 2, pressure had a significant effect on the total number of colonies in the juice (*p* < 0.05), while the pressure time and cycles had no obvious influence. The CFU in the juice was lower than 2 log cfu/mL, which was in line with the safety standard [33]. Considering the bactericidal effect and equipment cost, 500 MPa for 20 min at one cycle (A3B2C1) was selected as the optimized condition for UHPS to reduce the growth and propagation of microorganisms as well as to extend the shelf life of XL7 juice.

During the process of UHPS treatment, the juice could be made sterile only by adjusting the pressure, holding time and pressure cycles, while the pressure temperature during UHP treatment was not obvious, which was consistent with the study of Dhenge, et al. [34]. UHPS allows for sterilization at room temperature without heating. Therefore, it could not only solve the shortcomings associated with thermal treatment but also reduce the energy consumption associated with heating and cooling [35].

The influence of different sterilisation methods on the bactericidal efficacy and quality of XL7 juice is shown in Table 3. No microbial indexes were detected in XL7 juice after UHPS and HS treatments, indicating that both had good bactericidal efficacy. TSS content in HS– and UHPS–treated XL7 juice decreased significantly (*p* < 0.05) compared to the control group. Titratable acid in HS–treated XL7 juice decreased, while it increased slightly in the UHPS–treated group compared to the NS group (*p* < 0.05). The ratio of solid acid in UHPS–treated juice was closer to that of the NS group.

After sterilization, the L* value decreased, while the a* and b* values increased (*p* < 0.05). The △E of UHPS– and HS–treated juice were 0.74 ± 0.05 and 0.95 ± 0.02, respectively. Compared with NS juice, the juice–browning degree increased after sterilization and there was no significant difference in soluble quinone content between these three groups. Furthermore, the content of 5–HMF in the UHPS– and HS–treated juice was significantly higher than that in the NS group (*p* < 0.05), suggesting that a nonenzymatic browning reaction occurred during sterilization. Compared with the NS group, the contents of polyphenols and flavonoids and the free radical scavenging ability of DPPH and ABTS in HS–treated juice significantly decreased (*p* < 0.05), while these bioactive components in UHPS–treated juice had no significant change (*p* > 0.05). This observation suggested that UHPS could better maintain the bioactive compounds in XL7 juice. The CSR of UHPS–treated juice was significantly decreased, while it increased significantly in HS–treated juice compared with the NS group (*p* < 0.05), which might be due to the change of particle size and distribution of XL7 juice under UHP treatment, increasing the particle–particle and particle–juice interaction, as well as the stability of XL7 juice. Hence, the observations suggested that UHPS treatment could better maintain the functional quality of XL7 juice, which was consistent with the study of [36]. The thermal treatment may reduce the content of phenolic compounds and affect the environment and microstructure of food substrates, thus reducing its antioxidant activity in a complex way.

### 3.2. Effects of Different Sterilization Methods on Sensory Quality of XL7 Juice during Storage

The effects of different sterilization methods on the sensory quality of XL7 juice during storage are shown in Figure 1. The sensory scores of XL7 juice stored at different temperatures showed a downward trend with increasing storage time. The sensory scores of UHPS– and HS–treated juices showed the fastest decline trend during 37 °C storage, while the juice exhibited the lowest decline trend of the sensory scores during 4 °C storage. Hence, it can be speculated that low–temperature storage could be an effective way to maintain the sensory quality of XL7 juice. In addition, at the end point of storage at the same temperature, the sensory scores of UHPS–treated juice were 5.20 ± 0.31 (37 °C), 5.45 ± 0.37 (27 °C), and 7.40 ± 0.32 (4 °C), whilst the sensory scores of HS–treated juice were 5.13 ± 0.41 (37 °C), 5.43 ± 0.32 (27 °C), and 6.70 ± 0.31 (4 °C). Therefore, under the same temperature storage conditions, the sensory quality of UHPS–treated juice was better than that of HS–treated juice. The sensory score of UHPS–treated samples stored at 4 °C is the highest among all groups.

### 3.3. Effects of Different Sterilization Methods on Microbial Indexes of XL7 Juice during Storage

The effects of different sterilization methods on microbial indexes of XL7 juice during storage are shown in Table 4. The CFU in UHPS–treated juice increased at 6–7 days during 37 °C storage. Bacteria and fungi can form spores. When they sense a harsh environment, they will go into dormancy until conditions become more favourable for germination. Herein, spores in a dormant state of sterilized XL7 juice were self–repaired at 37 °C. However, the CFU still met the safety standard of <2 log cfu/mL. In addition, the mould, yeast, and *Escherichia coli*. were not detected in both UHPS– and HS–treated juice during storage. High pressure or heating treatment can lead to changes in the cell membrane and morphology, biochemical reactions, and alteration in the genetic mechanism, which is responsible for microorganism inactivation. These effects vary with the type of microorganism and food composition [37]. Herein, both UHPS and HS treatments could effectively kill microorganisms while maintaining the edible safety of XL7 juice during storage.

### 3.4. Effects of Different Sterilization Methods on the Glyconic Acid Content of XL7 Juice during Storage

The effects of different sterilization methods on the glyconic acid content of XL7 juice during storage are shown in Figure 2. With the extension of storage time, the TSS of XL7 juice stored at different temperatures showed a significant decrease trend (*p* < 0.05), and titratable acid showed a significant increase trend (*p* < 0.05). This change may be due to the degradation of carbohydrates in the juice during storage and the generation of organic acids [38]. This decreased the TSS content while increasing the titratable acid content. Furthermore, sucrose hydrolysis and utilization of reducing sugar by microorganisms in XL7 juice increased organic acid production, thus reducing the TSS and pH value. The TSS retention rate of UHPS– and HS–treated juice at 4 °C was 90.64 and 88.04%, respectively, and the variation range of titratable acid was 0.63–0.71 and 0.50–0.60 g/L, respectively. The change rate of TSS and titratable acid of the juice stored at other temperatures was higher than that of the juice stored at 4 °C. Under the influence of TSS and titratable acid changes, the solid–acid ratio of XL7 juice during storage decreased significantly (*p* < 0.05). UHPS–treated juice better preserved TSS and titratable acid and the contents of TSS and titratable acid in the UHPS group were higher than those in the HS group during storage (*p* < 0.05). The glyconic acid content in UHPS–treated juice was closer to the original juice, while it decreased after HS treatment. The content of 5–HMF increased significantly after HS treatment. Maillard reaction occurred after heating the XL7 juice, which destroyed the glyconic acid content of the juice and caused nonenzymatic browning during sterilization. Generally, a browning reaction normally occurred at the end stage of fruit ripening. However, some studies indicated that the internal browning reaction occurred during exposure towards oxidative stress, temperature, and prolonged storage. Hence, this study demonstrated an increase in 5–HMF concentration in watermelon juice advent to increase temperature storage, resulting in an intense browning reaction.

### 3.5. Effect of Different Sterilization Methods on Colour Quality of XL7 Juice during Storage 

The influence of different sterilization methods on L* and ∆E values of XL7 juice during storage is shown in Figure 3. When the storage time increased gradually, the L* value of juice stored at different temperatures presented a significant decline (*p* < 0.05), while the value of ∆E increased significantly (*p* < 0.05). When the juice was stored at 37 °C, the colour changed the fastest. At the end point of UHPS– and HS–treated juice groups, the ∆E value is 4.39 ± 0.04 and 4.92 ± 0.02, respectively, showing visible changes, while at the end point of UHPS–treated juice group stored at 4 °C, the ∆E is 1.53 ± 0.05. The degree of colour change was minimal.

The effects of different sterilization methods on the browning degree, soluble quinone content, and 5–HMF of XL7 juice during storage are shown in Figure 4. With the gradual increase of storage time, the browning degree, soluble quinone content, and 5–HMF of XL7 juice stored at different temperatures showed a significant increase trend (*p* < 0.05), indicating that enzymatic and nonenzymatic browning occurred during storage, and the degree of browning increased. The browning degree of both UHPS– and HS–treated juice increased most rapidly when the storage temperature was 37 °C, which might be related to the appropriate temperature for enzymatic and nonenzymatic browning reactions. In addition, when UHPS–treated juice was stored at 37, 27, and 4 °C, 5–HMF ranged from 0.57 to 4.09, 0.57 to 3.34, and 0.57 to 1.62 mg/100 mL, respectively, and soluble quinone content ranged from 0.076 to 0.092, 0.076 to 0.086, and 0.076 to 0.080, respectively. When the HS–treated juice was stored at 37, 27, and 4 °C, the variation range of 5–HMF was between 1.04 and 7.51, 1.04 and 4.37, and 0.57 and 2.30 mg/100mL, respectively, and the soluble quinone content was 0.075–0.087, 0.075–0.081, and 0.075–0.078, respectively. The content of 5–HMF changed greatly during storage, indicating that a nonenzymatic browning reaction was more likely to occur in juice during storage. However, the PPO activity, which is conducive to enzymatic browning reaction, decreased after UHP or heating treatment. The optimized temperature for PPO activity was approximately 40 °C. The temperature for the UHPS treatment in this study is 50 °C, which could not be completely inactivated in all the PPO but only decreased its activity.

The correlation analysis results of the colour index are shown in Figure 5. The L* values of UHPS– and HS–treated XL7 juice were negatively correlated with the browning degree, whilst the a* and b* values were positively correlated with the browning degree (*p* < 0.01). This observation was consistent with the results of the colour profiles. There were significant positive correlations between 5–HMF, soluble quinone content, and browning degree in the two groups (*p* < 0.01), suggesting that the nonenzymatic browning reaction was the main cause of colour change in XL7 juice during storage.

The effects of different sterilization methods on the contents of polyphenols and flavonoids in XL7 juice during storage are depicted in Figure 6. With the increase in storage time, the contents of polyphenols, flavonoids, and the antioxidant capacity of XL7 juice stored at different temperatures decreased significantly (*p* < 0.05). This change may be due to the oxidative condensation reaction between polyphenol and dissolved oxygen during storage [39]. This reaction ultimately reduced their antioxidant capacities. At the end point of the test, the polyphenol and flavonoids retention rates of UHPS–treated XL7 juice at 37, 27, and 4 °C were 60.61, 67.18, and 81.96% and 48.67, 56.12, and 82.41%, respectively, while the corresponding polyphenol and flavonoids retention rates in HS–treated XL7 juice were 59.17%, 64.14%, and 79.78%, and 43.64, 53.96, and 80.79%. The DPPH scavenging abilities of UHPS–treated juice stored at 37, 27, and 4 °C were 22.06 ± 0.61, 24.99 ± 0.67, and 33.48 ± 0.21%, respectively, while the ABTS scavenging ability was 16.37 ± 1.04, 18.13 ± 0.24, and 23.33 ± 0.77%, respectively. The DPPH scavenging abilities of HS–treated juice at 37, 27, and 4 °C were 16.67 ± 0.34, 24.19 ± 0.82, and 29.63 ± 0.75%, respectively, whilst the ABTS scavenging capacities were 13.94 ± 0.24, 14.80 ± 0.33, and 19.21 ± 0.23%, respectively. These observations indicated that cryogenic storage of UHPS preserved the bioactive components and antioxidant capacity of XL7 juice to the highest extent. This is consistent with the study by Yuan, et al. [40]. Thermal processing may reduce the content of phenolic compounds and affect the microstructure of the food matrix, thus reducing its antioxidant activity.

### 3.6. Effects of Different Sterilization Methods on the Stability of XL7 Juice during Storage

The effects of different sterilization methods on the CSR of XL7 juice during storage are shown in Figure 4. An interesting change in the stability of XL7 juice after sterilization was observed. During the storage period, the CSR of both UHPS– and HS–treated juice stored at different temperatures increased significantly (*p* < 0.05). Phenols can react with proteins in the juice to form macromolecular substances. The aggregation and sedimentation of particulate matter increased the CSR of the juice [41]. At the end point of the storage, the CSRs of UHPS–treated juice were 3.97 ± 0.19, 3.18 ± 0.21, and 2.75 ± 0.10% at 37, 27, and 4 °C, respectively, while the corresponding CSRs of HS–treated juice were 6.31 ± 0.03, 5.66 ± 0.12, and 5.12 ± 0.17% respectively. The increase rate of CSR was slower at low temperature during storage, and the CSR of HS–treated juice was higher than that of UHPS–treated juice, showing low stability. The CSR of HS–treated XL7 juice was the highest, followed by that in the NS juice, and that in the UHPS–treated juice was the lowest. The large particles in the juice were broken into smaller suspended particles under the influence of pressure. The HS treatment induced the solidification of proteins in the juice and destroyed the stability of the juice [42].

Collectively, cotreatment of low temperature with UHPS treatment might be a promising method to maintain the quality of XL7 juice. Both UHPS and HS treatments could maintain the microbial safety of XL7 juice in all tested samples. Szczepańska, et al. [43] treated apple juice with 450 and 600 MPa for 5 min, revealing that the apple juice could maintain microbial safety when stored at 4 °C for 12 weeks. During the low temperature (4 °C) storage, the bioactive compounds in XL7 juice could be maintained to a larger extent and had better stabilities when compared to the high temperature. Guan, et al. [44] treated mango juice with UHP treatment, finding that mango juice could be stored at 4 °C for 60 days. UHP treatment had no significant effect on the physicochemical properties of mango juice but caused little damage to bioactive components and antioxidant capacity. These findings showed that UHP treatment could better maintain the storage quality of fruit juice.

### 3.7. Prediction of Shelf Life of XL7 Juice Treated by Different Sterilization Methods

The shelf life of XL7 juice was predicted based on the Q10 model. The end points of the shelf life of UHPS– and HS–treated XL7 juice were 7, 14, and 12 days at 37, 27, and 4 °C, respectively, during storage. Therefore, the predicted shelf life of UHPS– and HS–treated juice was 68 and 41 days at 4 °C, respectively, during storage according to the formula. Therefore, UHP treatment with low–temperature storage could be an effective way to improve the shelf life of XL7 juice. The shelf life of UHPS–treatment XL7 juice stored at 4 °C was higher than that of HS–treated XL7 juice, which is consistent with the study of Lou, et al. [45]. The results in this study showed that the shelf life of the UHPS–treated XL7 juice was 68 d during storage, compared with the FC pear juice (319 d at 20 °C) and blueberry juice (225 d at 4 °C) in previous studies. The XL7 pear juice produced in this study had a shorter shelf life. However, the results were similar to those of old Chinese melon juice (27.5 d at 20 °C and 72 d at 4 °C). This might be because the shelf life was related to the types of raw materials of the juice. In this study, the shelf life of XL7 juice was preliminarily predicted by sensory quality and microbial index. In our future study, a dynamic model could be established to simulate the shelf life of XL7 juice by specific quality changes.

### 3.8. Changes of Volatile Components in XL7 Juice with Different Sterilization Methods before and after Storage

The fingerprint of volatile components of XL7 juice before and after storage by different sterilization methods is shown in Figure 7. The volatile components changed significantly during storage. A total of 48 volatile components were found in XL7 juice before and after storage, including 10 esters, 12 aldehydes, 10 alcohols, 7 ketones, 4 acids, 2 ethers, 1 hydrocarbon, 1 furan, and 1 pyrazine. The relative peak intensity of volatile components in XL7 juice before and after storage is displayed in Figure 5B. Ketones in UHPS–treated XL7 juice increased significantly compared with the NS group, which was due to the oxidation of alcohols and the increase of ketones, among which the largest increase was methyl heptenone, which had a green citrus–like smell. The types and concentrations of aroma components in HS–treated juice decreased significantly, indicating that heating may destroy aroma substances in juice, while UHPS could better preserve flavour substances in XL7 juice. Alcohols, esters, aldehydes, and other flavour substances in UHPS–treated juice stored at 27 and 4 °C increased to varying degrees. UHP could maintain or even activate the enzyme activity, decompose the large molecules in the juice, and release the small molecules of aroma substances to increase the detected volatile compounds. The peak intensity of aldehydes and ketones in UHPS–treated juice stored at 37 °C was lower than that of the UHPS–treated juice before storage, which may be due to the fact that the higher storage temperature promoted the growth of microorganisms and the increase of acidity led to the denaturement of enzymes, decreasing the concentration of aldehydes and ketones as well as the degree of aroma retention. The concentrations of alcohols, alkenes, pyrazines, and furans in the HS–treated juice decreased during storage, while other flavours increased. This may be because after the juice was heated, alcohols reacted with oxygen and organic acids to generate esters, aldehydes, ketones, and ethers [46].

## 4. Conclusions

Comparing the sterilization effect and quality of the optimized UHPS with HS treatment, it was found that the two sterilization methods could effectively ensure the microbial safety of XL7 pear juice. UHPS–treated XL7 juice could better maintain the quality of juice on the basis of keeping its edible safety, which would be a promising method of pear juice sterilization. UHPS–treated pear juices are not yet available in the market currently. This study presents high industrial relevance as it provides useful information related to the processing conditions that can allow for obtaining safe pear juices with prolonged shelf life. However, from the perspective of industrial applications, studies based on predictive microbiology and a deeper comprehension of the action mechanisms of UHPS on the physiological state of microbial cells still need to be investigated in our future work.

## Figures and Tables

**Figure 1 foods-12-02729-f001:**
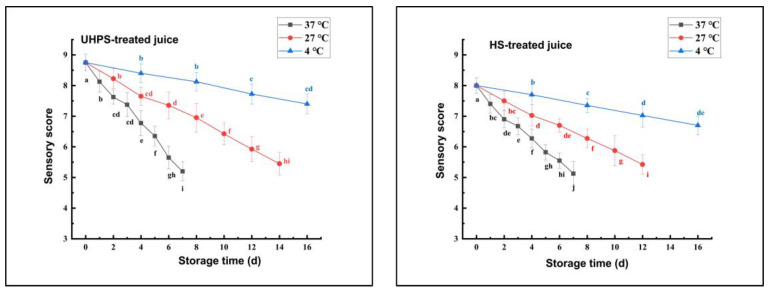
The sensory score of UHPS– and HS–treated Xinli No. 7 pear juice at 37, 27, and 4 °C during storage. Abbreviations: UHPS = ultra–high–pressure sterilization; HS = heating. Different letters represent there is significant statistical difference (*p* < 0.05) in sensory scores between different storage times under the same temperature.

**Figure 2 foods-12-02729-f002:**
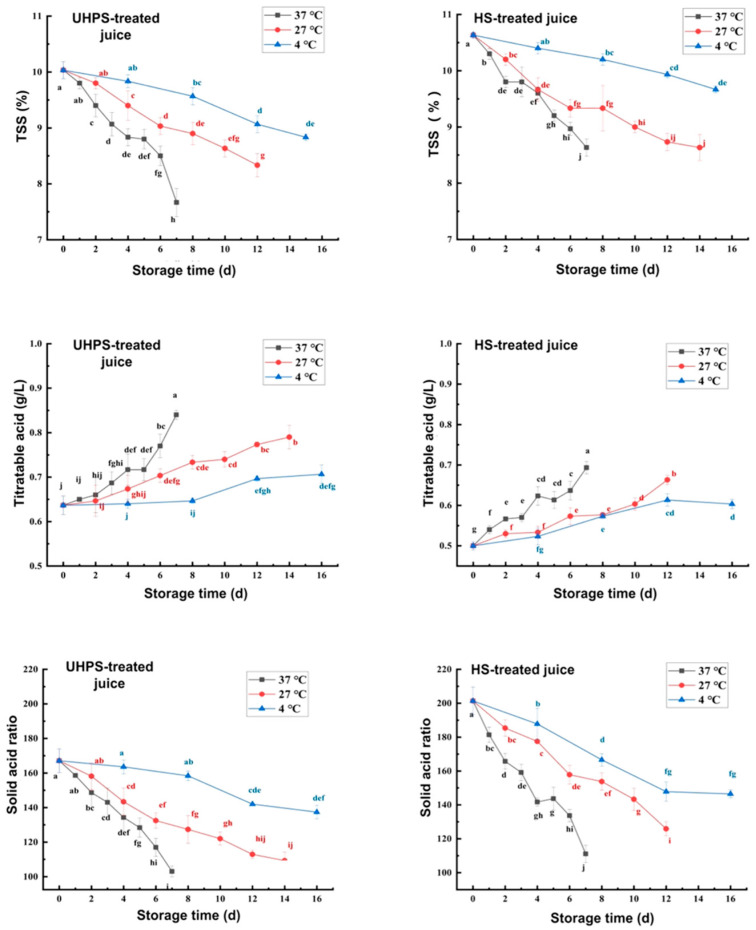
The total soluble solid content (TSS, %), Titratable acid (g/L), and solid–acid ratio of UHPS– and HS–treated Xinli No. 7 pear juice at 37, 27, and 4 °C during storage. Abbreviations: UHPS = ultra–high–pressure sterilization; HS = heating sterilization. Different letters represent there is significant statistical difference (*p* < 0.05) in TSS, titratable acid and solid acid ratio between different storage times under the same temperature.

**Figure 3 foods-12-02729-f003:**
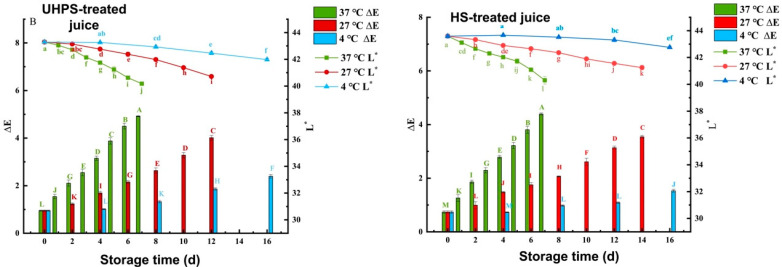
The colour difference (△E). Different lowercase and uppercase letters represent there is significant statistical difference (*p* < 0.05) in △E and L* value between different storage times under the same temperature, respectively.

**Figure 4 foods-12-02729-f004:**
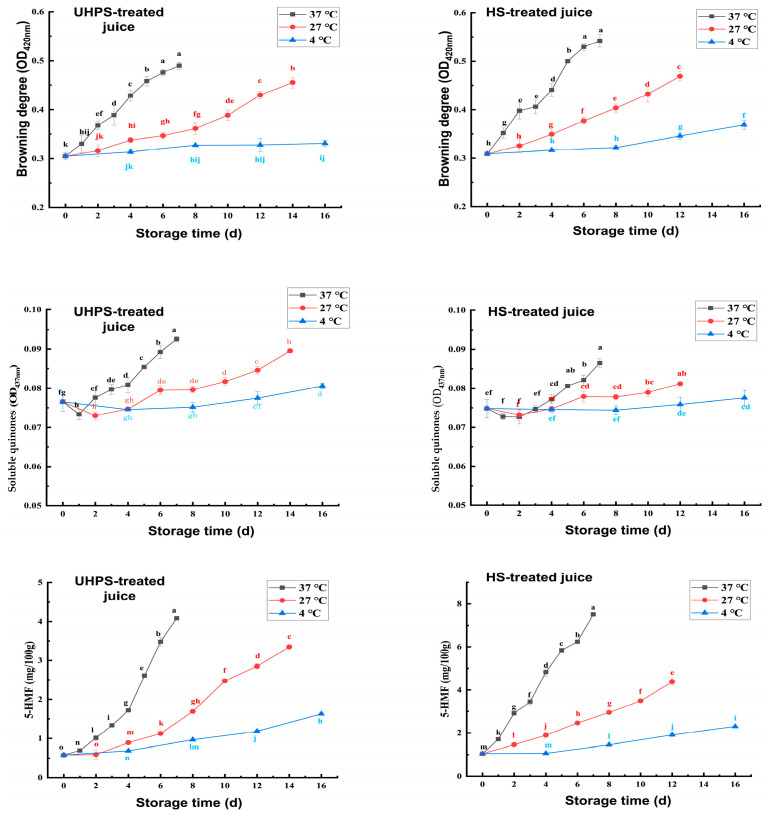
Browning degree, soluble quinone content, and 5–hydroxymethylfurfural (5–HMF) content of UHPS– and HS–treated Xinli No. 7 pear juice at 37, 27, and 4 °C during storage. Abbreviations: UHPS = ultra–high–pressure sterilization; HS = heating sterilization. Different letters represent there is significant statistical difference (*p* < 0.05) in browning degree, soluble quinone content, and 5–HMF between different storage times under the same temperature.

**Figure 5 foods-12-02729-f005:**
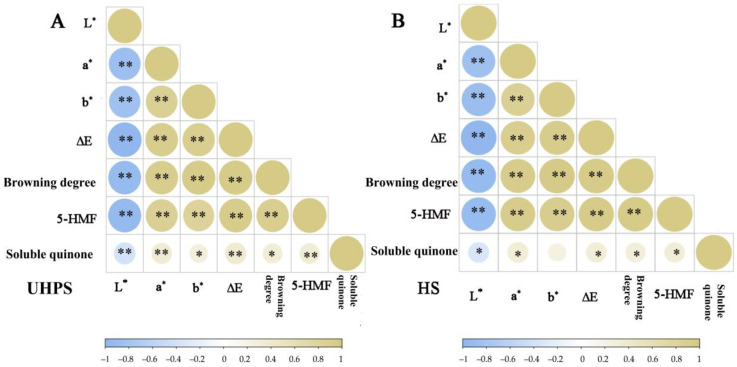
The correlation between colour and quality of UHPS– and HS– treated Xinli No. 7 pear juice at 37, 27, and 4 °C during storage. Abbreviations: UHPS = ultra–high–pressure sterilization; HS = heating sterilization. * means *p* < 0.05; ** means *p* < 0.01.

**Figure 6 foods-12-02729-f006:**
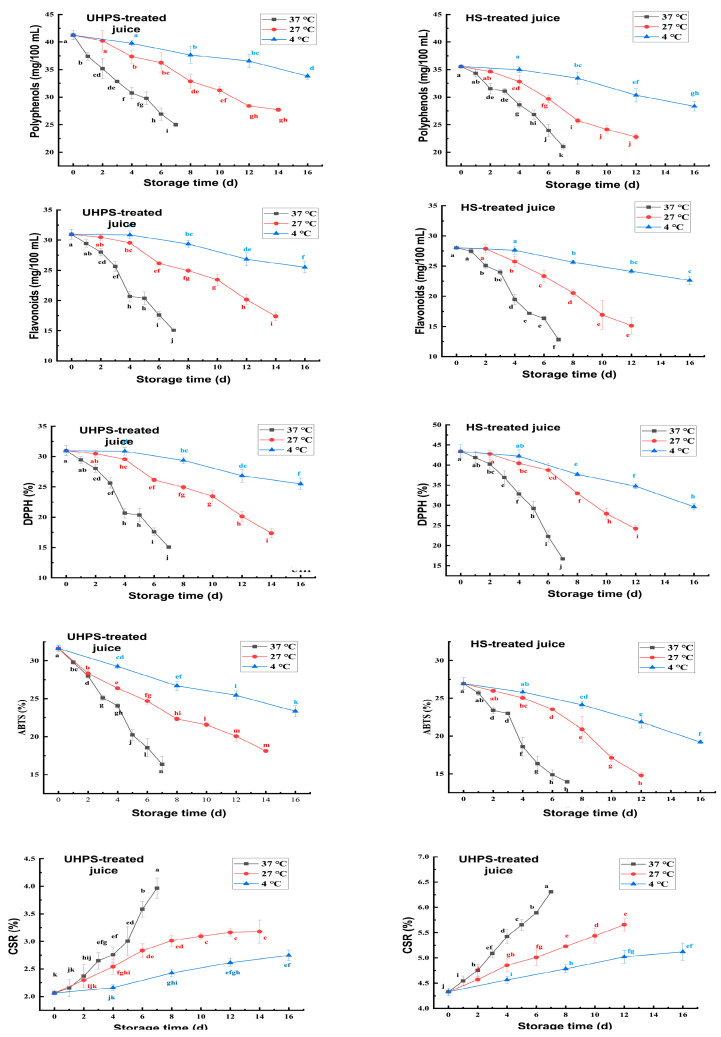
The polyphenols and flavonoids contents, DPPH, and ABTS scavenging capacities, and centrifugal sediment rate (CSR) of UHPS– and HS–treated Xinli No. 7 pear juice at 37, 27, and 4 °C during storage. Abbreviations: UHPS = ultra–high–pressure sterilization; HS = heating sterilization. Different letters represent there is significant statistical difference (*p* < 0.05) between different storage times under the same temperature.

**Figure 7 foods-12-02729-f007:**
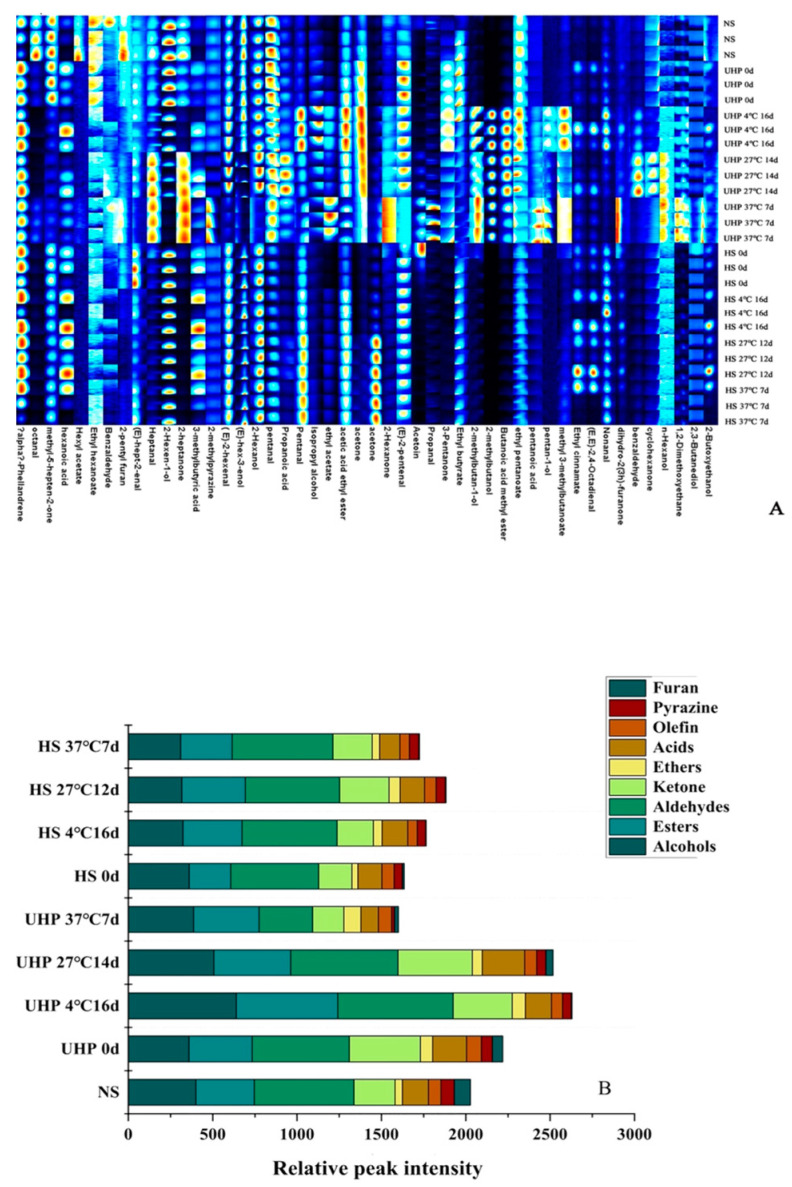
The fingerprint (upper) and relative peak intensity (lower) of volatile components of UHPS– and HS–treated Xinli No. 7 pear juice before and after storage at 37, 27, and 4 °C. Abbreviations: UHPS = ultra–high–pressure sterilization; HS = heating sterilization.

**Table 1 foods-12-02729-t001:** Orthogonal test.

Test Number	Pressure/Mpa (A)	Pressure Time/Min (B)	Cycle (C)	Column	CFU (lgcfu/mL)
1	1 (300 MPa)	1 (15 min)	1 (1)	1	3.14
2	1	2 (20 min)	2 (2)	2	2.22
3	1	3 (25 min)	3 (3)	3	1.40
4	2 (400 MPa)	1	2	3	0.40
5	2	2	3	1	0.27
6	2	3	1	2	0.45
7	3 (500 MPa)	1	3	2	–
8	3	2	1	3	–
9	3	3	2	1	–
K_1_	6.76	3.54	3.59	3.41	
K_2_	1.12	2.49	2.62	2.67	
K_3_	0.00	1.85	1.67	1.80	
k_1_	2.25	1.18	1.20	1.14	
k_2_	0.37	0.83	0.87	0.89	
k_3_	0.00	0.62	0.56	0.60	
Range	2.25	0.56	0.60	0.54	

Note: – represents no detected. K depicts the column number of orthogonal test, while the number means the test number.

**Table 2 foods-12-02729-t002:** ANOVA results of the orthogonal test.

Factor	Sum of Square	Degree of Freedom	Mean Square	F Value	*p* Value	Significance
Pressure (A)	8.751	2	4.376	20.213	0.047	*
Pressure (B)	0.485	2	0.243	1.121	0.471	
Cycle (C)	0.614	2	0.307	1.419	0.413	
Error	0.433	2	0.216			
Sum	17.183	9				

Notes: * represents there is statistical difference (*p* < 0.05).

**Table 3 foods-12-02729-t003:** Effect of different sterilization treatments on the sterilization effect and juice quality of XL7 juice.

	Quality Indexes	NS	UHPS	HS
Microbiological indicators	Moulds and yeasts (lgcfu/mL)	3.99 ± 0.03 ^a^	ND	ND
	CFU (lgcfu/mL)	4.33 ± 0.03 ^a^	ND	ND
	*E. coli* (lgcfu/mL)	2.4 ± 0.11 ^a^	ND	ND
Glyconic content	TSS (%)	10.97 ± 0.42 ^a^	10.63 ± 0.06 ^a^	10.03 ± 0.15 ^b^
	Acid (g/L)	0.63 ± 0.02 ^a^	0.64 ± 0.02 ^a^	0.50 ± 0.01 ^b^
	Solid acid ratio	17.36 ± 1.11 ^b^	16.71 ± 0.69 ^b^	20.14 ± 0.81 ^a^
Colour	L*	44.10 ± 0.32 ^a^	43.59 ± 0.15 ^b^	43.29 ± 0.06 ^b^
	a*	1.01 ± 0.06 ^b^	1.40 ± 0.16 ^a^	1.32 ± 0.06 ^a^
	b*	3.80 ± 0.18 ^b^	4.08 ± 0.16 ^a,b^	4.17 ± 0.05 ^a^
	△E	/	0.74 ± 0.05 ^a^	0.95 ± 0.02 ^b^
	Browning degree	0.27 ± 0.01 ^a^	0.31 ± 0.01 ^b^	0.31 ± 0.02 ^b^
	5–HMF	0.46 ± 0.01 ^c^	0.58 ± 0.04 ^b^	1.04 ± 0.04 ^a^
	Quinone content (ΔOD_437 nm_/g)	0.08 ± 0.01 ^a^	0.08 ± 0.00 ^a^	0.07 ± 0.00 ^a^
Functional properties	Polyphenols content (mg/100 mL)	41.21 ± 2.00 ^a^	41.25 ± 0.65 ^a^	35.53 ± 0.13 ^b^
	Flavonoids (mg/100mL)	32.46 ± 0.70 ^a^	30.96 ± 0.66 ^b^	28.02 ± 0.19 ^c^
	DPPH (%)	46.71 ± 1.18 ^a^	47.52 ± 0.27 ^a^	43.38 ± 1.64 ^b^
	ABTS (%)	32.41 ± 0.32 ^a^	31.61 ± 0.41 ^a^	26.92 ± 0.77 ^b^
Stability index	CSR (%)	3.73 ± 0.15 ^b^	2.06 ± 0.03^c^	4.33 ± 0.08 ^a^

Values = mean ± standard deviation (SD, *n* = 3). Values in the same row with different letters mean there is a significant difference between groups. NS = no sterilization; UHPS = ultra–high–pressure sterilization; HS = heating sterilization; ND = none was detected.

**Table 4 foods-12-02729-t004:** Effect of different sterilization treatments on microbiological indicators of XL7 juice during storage.

Microbiological Indicators	Storage Time (d)	Ultra–High–Pressure Sterilization (UHPS)			Heating Sterilization (HS)		
		37 °C	27 °C	4 °C	37 °C	27 °C	4 °C
CFU (log cfu/mL)	0	ND	ND	ND	ND	ND	ND
	1	ND	/	/	ND	/	/
	2	ND	ND	/	ND	ND	/
	3	ND	/	/	ND	/	/
	4	ND	ND	ND	ND	ND	ND
	5	ND	/	/	ND	/	/
	6	0.16 ± 0.01	ND	/	ND	ND	/
	7	0.34 ± 0.03	/	/	ND	/	/
	8	/	ND	ND	/	ND	ND
	10	/	ND		/	ND	/
	12	/	ND	ND	/	ND	ND
	14	/	ND		/	ND	/
	16	/	ND	ND	/	/	ND
Moulds and yeasts (log cfu/mL)	0	ND	ND	ND	ND	ND	ND
	1	ND	/	/	ND	/	/
	2	ND	ND	/	ND	ND	/
	3	ND	/	/	ND	/	/
	4	ND	ND	ND	ND	ND	ND
	5	ND	/	/	ND	/	/
	6	ND	ND	/	ND	ND	/
	7	ND	/	/	ND	/	/
	8	/	ND	ND	/	ND	ND
	10	/	ND		/	ND	/
	12	/	ND	ND	/	ND	ND
	14	/	ND		/	ND	/
	16	/	ND	ND	/	/	ND
*E. coli* (log cfu/mL)	0	ND	ND	ND	ND	ND	ND
	1	ND	/	/	ND	/	/
	2	ND	ND	/	ND	ND	/
	3	ND	/	/	ND	/	/
	4	ND	ND	ND	ND	ND	ND
	5	ND	/	/	ND	/	/
	6	ND	ND	/	ND	ND	/
	7	ND	/	/	ND	/	/
	8	/	ND	ND	/	ND	ND
	10	/	ND		/	ND	/
	12	/	ND	ND	/	ND	ND
	14	/	ND		/	ND	/
	16	/	ND	ND	/	/	ND

ND means none was detected. “/” means they were not detected under the corresponding condition.

## Data Availability

Data is contained within the article.

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
