# Peer review of "Evaluation of Ultra–High–Pressure Sterilization in Terms of Bactericidal Effect, Qualities, and Shelf Life of ‘Xinli No. 7’ (Pyrus sinkiangensis) Pear Juice"

_foods, 2023, doi:10.3390/foods12142729_

Round 1
Reviewer 1 Report
This study evaluated the efficiency of ultra-high pressure sterilization on bactericidal effect as well as some quality parameters of pear juice. Overall, it is a good study with practical value in food preservation. The research design and presentation of the manuscript are acceptable. The following points need attention during the revision
1. Provide reference for the estimation of glyconic acid content (sec. 2.8 in materials and methods)
2. Though the methodology is mentioned with previous references, I suggest the authors to add a brief description for important parameters.
Reviewer 2 Report
foods-2460527-peer-review-v1
Evaluation of ultra-high-pressure sterilization in terms of bac-2 tericidal effect, qualities and shelf-life of ‘Xinli No.7’ (Pyrus sinkiangensis) pear juice
Some suggestions are given below:
1- Introduction
Additional bibliographical references should be incorporated for a better support of some paragraphs.
Lines 27-28, page 1 “Pear is known as "the family of hundreds of fruits", with rich nutritional components 27 and medicinal value.”
Please include reference
Lines 28-29, page 1
“However, 29 NFC pear juice application study is few, and some issues related to the production and 30 maintenance of the pear juice still exist”
Please include reference
Lines73-76, page, These studies have been previously published,?
Please include a reference or be more specific
2- Materials and methods
Please revise the name 1,1-diphenyl-2-trinitrophenylhydrazine (DPPH).
3- Section 2.2. Operation of ultrasonic-ascorbic acid compound colour protection technology
Lines 104-105, page 3
“The ultrasonic power was set as 300 W for 10 min according to our preliminary study results.”
Please include a reference or be more specific
4- Section 2.8. Evaluation of glyconic acid content in XL7 juice
Please include a reference to method
5- Section 2.11. Identification of volatile components in XL7 juice
Lines 190-191, page 5, the authors write “The conditions of HS-190 GC-IMS instrument are shown in Table S1.”
Please include after section Conclusion
A Section Supplementary Material: The following supporting information can be downloaded at: www.mdpi.com/xxx/s1, Figure S1: title; Table S1: title; Video S1: title
6- Section 2.12. Sensory analysis of XL7 juice
The overall sensory evaluation score of XL7 juice was the average score of the 1 four indicators, as shown in Table S2.
Please include after the conclusions, the section corresponding to supplementary material, as the journal format suggests.
Also include tables S1 and S2, as an additional file, as suggested by the instructions for the author of this journal and others from MDPI.
A Section Supplementary Material: The following supporting information can be downloaded at: www.mdpi.com/xxx/s1, Figure S1: title; Table S1: title; Video S1: title.
When the manuscript under evaluation is accessed, only the paper file can be downloaded, the supplementary material is not observed.
7- Lines 379-395.
This paragraph should be supported by relevant bibliographical references, as it is presented it is weak and lacks interest.
The work after the suggestions made, could be considered again for its potential publication.
Moderate editing of English language required
Reviewer 3 Report
* Organization of whole paper from abstract to conclusion need to be improved.
* There are several typo mistakes and grammatical issues throughout the manuscript.
* Methods used in this manuscript are not clearly implemented.
* Results and discussion portion needs more comprehensive explanations based on study point.
* Tables should be revised in an articulated way.
Consider these papers in your manuscript:
Maturity evaluation of supply chain procedures by combining SCOR and PST models
Using fuzzy DEMATEL and fuzzy Similarity to develop a human capital evaluation model
Mathematical modeling of Green closed loop supply chain network with consideration of supply risk: Case Stud
* Organization of whole paper from abstract to conclusion need to be improved.
* There are several typo mistakes and grammatical issues throughout the manuscript.
* Methods used in this manuscript are not clearly implemented.
* Results and discussion portion needs more comprehensive explanations based on study point.
* Tables should be revised in an articulated way.
Consider these papers in your manuscript:
Maturity evaluation of supply chain procedures by combining SCOR and PST models
Using fuzzy DEMATEL and fuzzy Similarity to develop a human capital evaluation model
Mathematical modeling of Green closed loop supply chain network with consideration of supply risk: Case Study
Author Response
Please see the attachment

Reviewer 4 Repor
The present study demonstrates the investigation of ultra-high-pressure sterilization (UHPS), on the physicochemical, biochemical and sensory properties of Xinli No.7 pear juice during processing and storage under different conditions. The manuscript was well prepared and written with a clear English language level. In my humble opinion, however, several minor suggestions as indicated below should be clarified and revised.
· Table and figure ordering system: I would recommend running the numbers continuously, i.e. Table 1, 2, 3 …n, not as Table 1a, 1b…
· Line 48: Both pasteurization and sterilization could ensure the safety of products.
· Line 99-105: If applicable, a reference regarding this treatment should be addressed.
· Line 187-191: If applicable, a reference to the condition applied in HS-GC-IMS measurement should be mentioned.
· Line 240: Results and discussion?
· Table 2b: The data presentation is not that easy to understand. Please kindly revise.
· Figure 1 should be presented in a consistent manner with Figure 2 and 3.
· Line 291: what would be the reason linked to physical and chemical changes of product regarding this statement.
· Line 303-305: The inhibitory effect of UHPS on microbial cells and spores should be additionally discussed.
· Line 315-317: If applicable, a reference regarding this treatment should be addressed.
· Line 359-361: How possible that the enzymes were not completely inactivated by the concept of sterilization. Please provide a discussion on this observation.
· Collectively, the impact of storage temperature on physicochemical and biochemical changes of juice products should be extensively discussed.
· Line 382-385: If applicable, a reference regarding this treatment should be addressed.
· Line 401-403: If applicable, a reference regarding this treatment should be addressed.
· Line 422-423: How could the reduction in molecular size of polyphenols affect their bioactivities? Please provide a brief discussion on this topic.
· Line 465-467: It seems that this statement is in contradiction with the results in Table 2b.
· Line 470-472: If applicable, a reference regarding this treatment should be addressed.
· Line 479: limitations or drawbacks of UHPS found in this study should be also mentioned.
The manuscript was well prepared and written with good English language level.
Reviewer 5 Report
Line 31: The letter b in browning should be small.
Line 34: Delete as.
Line 45: corruption is not the right term. Change it to contamination.
Line 46-47: Reframe the sentence using proper English.
Line 50: change addressing to address.
Line 51: change reducing to reduce.
Line 52: change is to apply to utilizes. Please mention the numerical value for the pressure used in UHPS.
Line 78: Expand CFU. Instead of cfu it would be better to use the term microbiological quality.
Section 2.3.2 to section 2.3.5 could be merged into a single subsection. The title of sub-title could be “Effect of processing parameters on the number of XL7 juice colonies”.
Line 131: Mention the correct table number.
Line 136: “put it in a” change it to scientific English.
Line 173: Mention the standard curve equation with R2 values.
Line 184: Mention the standard curve equation with R2 values.
Line 198: thestandard change to the standard
Line 200: lg change to log
Line 205: Change title to accelerated shelf-life test
Line 232: 0 d change to 0 days
Line 245: as per safety standard of which regulatory agency.
Line 247: as well as to extend
Line 257: were significantly decreased change to decreased significantly
Line 259-262: There are instances where authors have used “were decreased”. Instead of this only decreased could be used.
Line 263: There is no point of including these words “which 263 could not be observed by naked eye”. Delete it.
Line 268: Browning change B to b.
Line 268: was occurred change to occurred
Line 268-272: Break it into small sentences for better understanding.
Line 303-305: “The CFU in UHPS-treated juice increased at 6-7 days 303 during 37 ℃ storage, which might be because the spores in dormant state of sterilized XL7 304 juice were self-repaired at 37 ℃.” Reframe this sentence with usage of proper terminology for better understanding.
Line 382: were significantly decreased is incorrect. Correct it.
Line 382-384: Mention the related reference.
Line 401-404: Mention the related reference.
Line 441-442: Incorrect sentence. and reduced and the raw materials.
The whole manuscript is having grammatical mistakes, which needs critical attention. The style of writing need to be improved for better understanding of the readers. There are lot of big sentences, which can be written into smaller sentences.
Round 2
Reviewer 2 Report
foods-2460527-peer-review-v2
The authors have considered the suggestions made in the first review, however one of them should be reviewed again. I apologize, in case I have not been clear in the wording of it.
Q8.Identification of volatile components in XL7 juice. Lines 190-191, page 5, the authors write“The conditions of HS-190 GC-IMS instrument are shown in Table S1.”Please include after section Conclusion
A8. The supplementary tables were provided after conclusion section.
The supplementary material is not uploaded according to the suggestions of MDPI Journals
Supplementary material (Tables, Figures) should be presented according to the Journal's suggestions.
"The material should be uploaded as an additional file in word or pdf format"
“Only, a brief paragraph where it is mentioned that the article has supplementary material should be incorporated after the conclusions.”
Please see the following recently published article for the proper presentation of the supplementary material.
https://www.mdpi.com/2304-8158/12/13/2476
Foods 2023, 12(13), 2476; https://doi.org/10.3390/foods12132476 (registering DOI)
Example
Supplementary Materials: The following supporting information can be downloaded at: https:
//www.mdpi.com/article/10.3390/foods12132476/s1, Table S1: Samples of geo-authentic Chinese
yam; Table S2: Samples of non-authentic Chinese yam.
After the suggested change, the manuscript should be accepted for publication.
Moderate editing of English language required
Reviewer 3 Report
Consider these papers in your manuscript:
Maturity evaluation of supply chain procedures by combining SCOR and PST models
Using fuzzy DEMATEL and fuzzy Similarity to develop a human capital evaluation model
Mathematical modeling of Green closed loop supply chain network with consideration of supply risk: Case Study
Consider these papers in your manuscript:
Maturity evaluation of supply chain procedures by combining SCOR and PST models
Using fuzzy DEMATEL and fuzzy Similarity to develop a human capital evaluation model
Mathematical modeling of Green closed loop supply chain network with consideration of supply risk: Case Study
